# stLaBraM: Interpretable EEG Foundation Model with Factorized Spatial-Temporal Patch Embedding for Self-Supervised Learning

## Abstract

Electroencephalography (EEG) foundation models aim to learn robust and transferable representations from large-scale EEG datasets, which are essential for enabling clinical and cognitive applications, such as rapid neurological screening, seizure detection, and brain state decoding. Current architectures struggle to combine interpretability and high performance for self-supervised masked modeling of EEG signals. In medical contexts, interpretability is especially important, because transparent models foster trust and facilitate clinical adoption. In this work, we introduce novel, interpretable spatial and temporal filters in the patch embedding module, advancing EEG foundation models and outperforming the previous state-of-the-art LaBraM architecture. We demonstrate that our approach significantly reduces reconstruction loss during self-supervised pre-training, enhancing the performance of the masked language model (MLM). Our new model outperforms the original LaBraM in standard EEG classification benchmarks and offers unique insights into the second-order dynamical properties and cortical locations of neuronal sources pivotal for self-supervised masked modeling. These results position stLaBraM as a compelling foundation model for EEG, advancing both performance and interpretability in self-supervised neurophysiological representation learning.

## 1 Introduction

Automated analysis of electroencephalography (EEG) data is pivotal for advancing large-scale neuroscience and clinical research, with applications from cognitive state decoding and biomarker discovery to neurological screening, seizure detection, and brain–computer interfaces (Abdulkader et al., 2015; Craik et al., 2019; Yuan et al., 2024). Accessibility of open-source EEG processing toolkits—such as MNE-Python (Gramfort et al., 2013) for preprocessing and TorchEEG (Zhang et al., 2024) for deep learning pipelines—has accelerated progress in this domain.

Recent years have witnessed the emergence of EEG foundation models, which employ large-scale, predominantly self-supervised pre-training to learn generalizable and reusable neural representations. Prominent examples include BrainWave (Yuan et al., 2024), a transformer-based model utilizing masked modeling for clinical EEG decoding; CBraMod (Wang et al., 2025), using a criss-cross architecture to promote cross-dataset invariance; and GREEN (Paillard et al., 2025), which leverages learnable wavelets and Riemannian geometry for interpretable biomarker identification. While these approaches have led to notable performance gains, a critical gap remains: most current models do not adequately provide interpretable representations, which are essential for clinical trust, scientific understanding, and regulatory acceptance. In addition, there is a growing recognition that model architectures tailored to the distinctive characteristics of biomedical data can substantially improve the quality and relevance of learned embeddings. Rather than adopting generic solutions, integrating domain-specific knowledge—such as neuroanatomical constraints or temporal dynamics—into model design offers the potential to produce representations that are both more informative and clinically meaningful.

A central challenge is the complex spatio-temporal structure of EEG signals and the need to align model representations with neurophysiological phenomena. Transformer-based architectures and patch-based masked auto-encoding (He et al., 2021; Chien et al., 2022)—as instantiated by architectures like LaBraM (Jiang et al., 2024)—have demonstrated strong self-supervised pre-training performance. However, their patch embedders rarely incorporate explicit neuroscientific priors, yielding latent features that are difficult to interpret in anatomical or functional terms. This limits their explanatory transparency and their integration into established neuroscience and clinical workflows (e.g., source localization and neuroscientific hypothesis validation).

According to (Biran & Cotton, 2017) : "Interpretability is the degree to which a human can understand the cause of a decision." Analysis of the explainable and interpretable EEG foundation model weights would support the subsequent use of electromagnetic inverse modeling (Spinelli et al., 2000) to locate neuronal populations whose activity appears pivotal in forming the latent representations. When working with time-resolved electrophysiological brain imaging modalities such as EEG or MEG, not only spatial but also dynamical properties of the neuronal sources are of critical interest (Buzsáki, 2006).

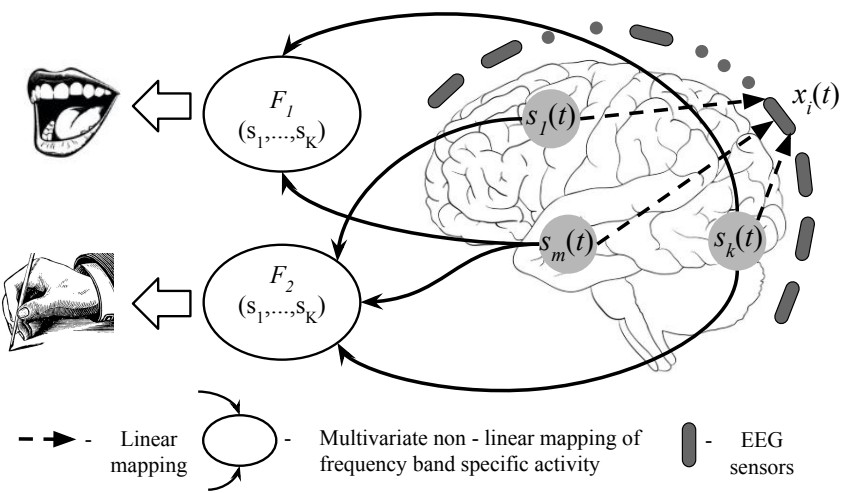

Figure 1: Simplified diagram illustrating linear mapping of neuronal sources to EEG sensors and fundamentally nonlinear relationships between the activity of brain sources and the behavior.

To address these challenges, we propose **stLaBraM**, an architecture that for the first time endows an EEG foundation model with explicit, rigorously interpretable factorized spatial and temporal filter banks. Drawing on classical signal processing and results of the existing literature, e.g. (Petrosyan et al., 2021), this design enables model branches to correspond to well-defined neural sources—for example, specific brain regions activated with rhythmic time series. The use of factorized spatial-temporal filters can be justified by Figure 1. Human behavior is undoubtedly a very complex and nonlinear function of the activity of neuronal populations, which justifies the use of deep learning models for EEG decoding. However, the EEG signal is simply a linear mixture of the activity of the local field electric potentials formed by the activity of neuronal sources (Mosher et al., 2002). Also, the activity of neuronal sources has characteristic dynamics whose second-order properties can be reflected using the power spectral density profiles (Pereira et al., 2021).

We evaluate stLaBraM on self-supervised pre-training and challenging downstream classification benchmarks, demonstrating state-of-the-art performance and interpretability. Our main contributions are:

- We propose a **novel patch embedding architecture** which, for the first time in an EEG foundation model, factorizes learnable spatial (topographic) and temporal (frequency band) filters—yielding codebooks interpretable in neurophysiological terms and enabling direct and rigorous mapping to brain sources and oscillatory processes.

- **Model-agnostic improvement:** Our interpretable patch embedding design leads to consistent accuracy gains when integrated with different EEG foundation model backbones—notably LaBraM and CBraMod—across pre-training, fine-tuning, and out-of-distribution (OOD) clinical evaluation.

- **State-of-the-art (SOTA) performance:** We consistently outperform previous models, including LaBraM and CBraMod, in masked modeling loss as well as balanced accuracy on TUAB (Obeid & Picone, 2016), ICD-10 (see Appendix A for dataset details), and event/cognitive datasets.

- We introduce a **principled, interpretability pipeline** for visualizing spatial and frequency patterns, permitting domain experts to directly inspect the neural features driving model predictions. When anatomical information in the form of MRI is available the pivotal sensor-space spatial patterns can be rigorously mapped to the cortical source-space.

## 2 BACKGROUND AND PROBLEM STATEMENT

EEG records potential difference on the scalp with a set of $M$ electrodes. At each time instance $t$ this results in a vector of sensor signals $\mathbf{x}(t) = [x_1(t), \ldots, x_M(t)]^\top$. The data can be modeled as a linear superposition of gain vectors $\mathbf{g}_i = \mathbf{g}(\mathbf{r}_i)$ corresponding to the $i$-th neuronal macro-population located at $\mathbf{r}_i$ scaled with time varying coefficients $s_i(t)$ called source time series and reflecting the electrical activity of the large conglomerates of spatially segregated neurons. This mixture is contaminated with noise vector $\mathbf{e}(t)$ that has a complex spatial-temporal correlation structure. Formally the generative equation can be written as

$$\mathbf{x}(t) = \sum_{i=1}^{N} \mathbf{g}_i s_i(t) + \mathbf{e}(t). \tag{1}$$

Gain vectors $\mathbf{g}_i$ can be visualized as sensor-space patterns distributed over scalp and inverse modeling procedures can be used to map them to the cortical source-space determine source locations $\mathbf{r}_i$. the neuronal sources and infer $\mathbf{r}_i$. Typically source time series $s_i(t)$ have distinct and physiologically meaningful second-order dynamical properties well described by the power spectral density (PSD) profiles $S_i(\omega)$. Deep neural networks for EEG typically begin with front-end feature extraction layers that learn spatial and/or temporal filters. When such filters are explicitly factorized into spatial and temporal parts, their parameters become interpretable under optimal filtering theory assumptions (Haufe et al., 2014; Petrosyan et al., 2021): spatial weights can be mapped to gain vectors, and temporal filters relate to the characteristic frequency bands of the neuronal sources.

Therefore, our goal is to endow a foundation model with interpretable initial layers comprising factorized spatial and temporal filtering operations. The obtained solution will not only be more trustworthy but will for the first time allow for discovery of the neuronal sources whose activity is pivotal for deriving the latent representations of EEG segments. When applied to downstream tasks, the spatial and spectral patterns derived from these filters can be compared between conditions to allow physiologically meaningful conclusions about differences in two EEG conditions.

## 3 STLABRAM: INTERPRETABLE PATCH EMBEDDING SOLUTION

LaBraM (Jiang et al., 2024) is a large-scale EEG foundation model leveraging channel patching, neural tokenization, and transformer pre-training for robust, generalizable representations across diverse EEG datasets and tasks. It utilizes a neural tokenizer trained on vector-quantized spectral features (amplitude and phase) of EEG channel patches, enabling masked prediction objectives and efficient self-supervised learning. However, the original patch embedding in LaBraM extracts temporal features independently for each channel, without explicit spatial mixing or leveraging frequency band information. This limits the model's neurophysiological interpretability and practical applicability in clinical or scientific settings.

To address these challenges, we design the patch embedding module to integrate:

1. **Spatial unmixing and Virtual Channels**: In the first stage, we combine the original EEG channels into virtual channels using the learnable weights. In other words the $k$-th *virtual*

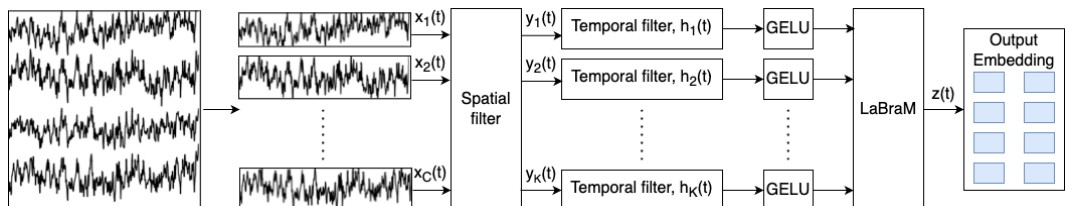

Figure 2: The proposed *stLaBraM* architecture incorporates parameterized spatial–temporal filtering modules within the patch embedder, supporting interpretability and robust representation learning.

*channel* is a linear combination of sensor signals with the learned weights. Formally, to derive the $k$-th virtual channel, $k = 1, \ldots, K = 16$ the virtual channel signal $y_k(t)$ at time $t$ is computed as:

$$y_k(t) = \sum_{i=1}^{C} w_{ki}\, x_i(t),$$

where $x_i(t)$ is the signal from the $i$-th electrode at time $t$ ($i = 1, \ldots, C$, with $C = 16$), and $w_{ki}$ are the learnable spatial filter weights. This enables the network to focus on the activity of specific neural populations.

2. **Frequency-Specific Temporal Filtering:** After spatial filtering, we obtain $K$ virtual channels, each represented as a time series $y_k(t)$ for $k = 1, \ldots, K$. For each virtual channel, we apply a bank of $K$ one-dimensional temporal convolutional filters $\{h_1, h_2, \ldots, h_K\}$, where each $h_k$ is initialized as a finite impulse response (FIR) filter of size $L$ targeting the specific EEG frequency band (e.g., delta, theta, alpha, etc.). This operation can be formally written as:

$$r_k(t) = y_k(t) * h_k(t) = \sum_{\tau=0}^{L-1} h_k(\tau)\, y_k(t - \tau),$$

where $*$ denotes convolution, $h_k(\tau)$ is the impulse response of the $k$-th temporal filter of length $L$, and $r_k(t)$ is the temporally filtered output for virtual channel $k$.

Stacking these outputs yields a set of $K$ band-filtered time series each corresponding to a specific neural population identified during the learning process.

During training, both the spatial filter weights $\mathbf{w}_k$ and temporal filter coefficients $h_k(t)$ are adaptively optimized to extract features relevant for pre-training or downstream tasks. By initializing temporal filters to cover physiologically meaningful frequency bands and jointly training all parameters end-to-end, the patch embedding module learns interpretable spatial and spectral patterns. These space-frequency filtered signals are then further processed by temporal convolutions and non-linearities to generate representations for masked modeling and classification.

**Interpretable spatial and frequency domain patterns visualization.** To support neuroscientific interpretation, we propose a visualization pipeline (Haufe et al., 2014) extended in this work to the simultaneously-trained spatial-temporal filtering case for analyzing the learned spatial $\mathbf{w}_i$ and temporal filters $h_i(t)$ (see Fig. 2). By leveraging the second-order spatial-temporal statistics of the data, we recover the generative topographies $\mathbf{g}_i$ and source power spectral density profiles (see Eq. 1).

The spatial and the temporal filters during training attempt to both tune to the target population and untune from the interfering signals. It is important to realize that these processes are taking places concurrently and therefore if the spatial filter has learned to ignore specific interfering signal, the temporal filter will not attempt to untune from it, and vice versa. Therefore, when computing spatial patterns one needs to take into account the data filtered in the specific frequency band and when calculating frequency domain pattern the appropriately spatially filtered and not the raw data need to be used.

Therefore, the spatial patterns $\mathbf{g}_i$ , $i = 1, \ldots, N$ corresponding to the spatial filters $\mathbf{w}$ can be recovered as:

$$\mathbf{g}_i = \sigma_{s_i}^{-2} \mathbf{R}_{x*h_i} \mathbf{w}_i \propto \mathbf{R}_{x*h_i} \mathbf{w}_i, \text{ where } \mathbf{R}_{x*h_i} = \mathbb{E}[(\mathbf{x}(t).*h_i(t))(\mathbf{x}(t).*h_i(t))^\top]. \tag{2}$$

In the above in order to compute topographies $\mathbf{g}_i$ of sources that help most in the pretraining or a downstream task we are using the data filtered with the corresponding temporal filter $h_i(t)$. Thus, each pair $(\mathbf{g}_i, \mathbf{w}_i)$ represents a generative topography and the corresponding spatial filter. While $\mathbf{g}_i$ reflects the generative structure in equation (1), $\mathbf{w}_i$ optimizes signal-to-noise extraction for decoding.

Similarly the power spectral density (PSD) profiles of sources can be found as

$$S_i(\omega) = PSD(\mathbf{w}_i^\top \mathbf{x}(t))|FFT(h_i(t))|.$$

## 4 DATASETS AND TRAINING PROTOCOLS

**Pre-training Dataset.** We constructed the pre-training set from the TUH EEG Corpus (Obeid & Picone, 2016), comprising over 3,600 hours of clinical EEG. To ensure uniformity and quality, all recordings were mapped to a standardized montage of 16 clinically relevant channels (FP1, FP2, F3, F4, F7, F8, C3, C4, P3, P4, O1, O2, T3, FZ, CZ, PZ); files missing any channel were excluded. Segments under 5 minutes were removed, and the first and last 60 seconds of every recording were discarded to minimize boundary effects. Signals were bandpass and notch filtered, downsampled to 200 Hz, and divided into 16-second non-overlapping windows. Artifactual epochs with extreme amplitudes were discarded.

**Downstream Evaluation (Fine-tuning).** For downstream classification, we evaluated on four benchmark datasets to assess transfer and generalization: (1) the TUAB Abnormal EEG Corpus (Obeid & Picone, 2016) for abnormal/normal clinical EEG classification, (2) the TUEV dataset (Obeid & Picone, 2016) for event-related abnormality detection, (3) the PhysioNet Motor Imagery (MI) dataset (Goldberger et al., 2000) for non-clinical cognitive classification, and (4) an independent multi-class clinical corpus labeled with ICD-10 diagnostic codes (see Appendix A) assigned by expert clinicians. For all datasets, we followed the same channel selection and preprocessing pipeline as used in pre-training: only recordings with all 16 standardized channels were included, data were bandpass and notch filtered, downsampled to 200 Hz, segmented into fixed-length windows (16 seconds), and artifact-laden epochs were discarded. Subject-level stratified splits were employed consistently to avoid data leakage between training and evaluation sets.

**Training Protocols.** All models were trained and evaluated under identical conditions to ensure a fair comparison. Each model was run across three independent trials, varying only in random parameter initialization while keeping all dataset splits, hyperparameters, and training procedures fixed. To assess robustness, performance was further evaluated on multiple resampled test subsets (1000 bootstraps). Complete details on hyperparameters, training schedules, and computational resources are provided in Appendix B.

## 5 RESULTS

### 5.1 PRE-TRAINING: MASKED MODELING PERFORMANCE

**Pre-training objective and loss interpretation.** Our pre-training setup employs a masked modeling paradigm, where the model is trained to infer the content or position of masked input patches using context from the visible patches. In this formulation, the model receives EEG sequences with a random subset of patches masked, and its task is to predict the correct positions (indices) of these masked patches based on the representations of the unmasked ones. This is implemented as a cross-entropy position classification loss:

$$\mathcal{L}_{\text{rec}} = -\sum_{i \in \mathcal{M}} \log p(\mathbf{y}_i = \hat{\mathbf{y}}_i \mid \mathbf{x}_{\backslash i}), \tag{3}$$

where $\mathcal{M}$ is the set of masked token indices, $\mathbf{y}_i$ is the true position label for patch $i$, and $\hat{\mathbf{y}}_i$ is the predicted position, conditioned on all unmasked observations $\mathbf{x}_{\backslash i}$. This encourages learning global and robust representations that enable the model to recover missing or occluded information.

To further regularize training and improve generalization, a symmetric version of the reconstruction loss is calculated over the unmasked (visible) patches as well. The total loss used to train the encoder is then

$$\mathcal{L}_{\text{total}} = \mathcal{L}_{\text{rec}} + \mathcal{L}_{\text{rec}}^{\text{sym}}, \tag{4}$$

where $\mathcal{L}_{\text{rec}}^{\text{sym}}$ is the cross-entropy loss computed analogously for the visible tokens. By optimizing reconstruction on both masked and unmasked parts of the input, the model is guided to learn more stable and transferable representations.

Table 1: Comparison of pre-training metrics for LaBraM and stLaBraM after 70 epochs. stLaBraM achieves higher MLM accuracy and lower reconstruction and total loss, with negligible computational overhead.

| Model | Params | MLM Acc | $\mathcal{L}_{\text{rec}}$ | $\mathcal{L}_{\text{total}}$ | Training time (hh:mm:ss) |
|---|---|---|---|---|---|
| LaBraM (baseline) | $9.154 \times 10^6$ | 0.259 | 1.701 | 6.804 | 1:44:31 |
| **stLaBraM (ours)** | $9.155 \times 10^6$ | **0.363** | **1.314** | **5.256** | 1:46:57 |

As shown in Table 1 and the loss/accuracy curves in Figure 3, stLaBraM achieves a significantly higher masked language modeling (MLM) accuracy and a substantially reduced reconstruction loss compared to baseline LaBraM (MLM acc: 0.363 vs 0.259, representing a 40.2% relative improvement, and $\mathcal{L}_{\text{rec}}$: 1.314 vs 1.701, a 22.8% relative reduction) at nearly every epoch, highlighting a consistent performance gap throughout training.

To evaluate the model-agnostic benefits of our approach, we additionally integrated interpretable spatial-temporal filtering into the CBraMod architecture. As detailed in Appendix C, similar trends are observed: stCBraMod consistently outperforms the baseline, yielding mean squared error (MSE) reduction by approximately 43.1%.

Importantly, these improvements are achieved with negligible additional computational overhead: both the number of parameters and the training time remain nearly unchanged for the st-augmented models. All models—including LaBraM, stLaBraM, CBraMod, and stCBraMod—were pre-trained from scratch on matched TUH splits using the masked modeling objective, with full training and evaluation protocols detailed in Appendix B. The persistent performance gap observed across both architectures further supports that our interpretable patch embedding leads to fundamentally improved training dynamics and model representations.

## 5.2 Downstream and Model-Agnostic Evaluation

Our interpretable front-end yields consistent, model-agnostic improvements across a diverse range of tasks. For example, on the TUAB benchmark, stLaBraM outperforms LaBraM with a balanced accuracy of $0.810 \pm 0.004$ versus $0.797 \pm 0.004$—a 1.6% relative accuracy gain—when using a patch embedding module frozen after pretraining. We freeze this module to preserve the general and interpretable spatial-temporal filters learned on large-scale data, enabling a fair assessment of their transferability to downstream settings. Similar trends persist across event-related and cognitive domains: on TUEV, balanced accuracy improves from 0.581 to 0.602 (a 3.6% relative gain), while on PhysioNet MI, accuracy substantially increases from 0.471 to 0.634 (a relative improvement of 34.6%), again with the pretrained patch embedder kept frozen, underscoring robust transferability to both out-of-distribution and non-clinical settings.

CBraMod benefits similarly: the stCBraMod variant delivers a statistically significant relative improvement of 2.3% over the baseline, further demonstrating that our spatial-temporal patch embedding enhances not only LaBraM but also other advanced EEG architectures (see Appendix C for detailed metrics).

Ablation studies stress the necessity of combining both spatial and temporal filtering: while accuracy shows negligible change for spatial-only architectures ($0.798 \pm 0.004$), the full spatial-temporal block yields a clear improvement ($0.810 \pm 0.004$). These analyses also show enhanced robustness to data corruption, highlighting that our method leverages physiologically meaningful structure. Additional results and ablations are presented in Appendix D.

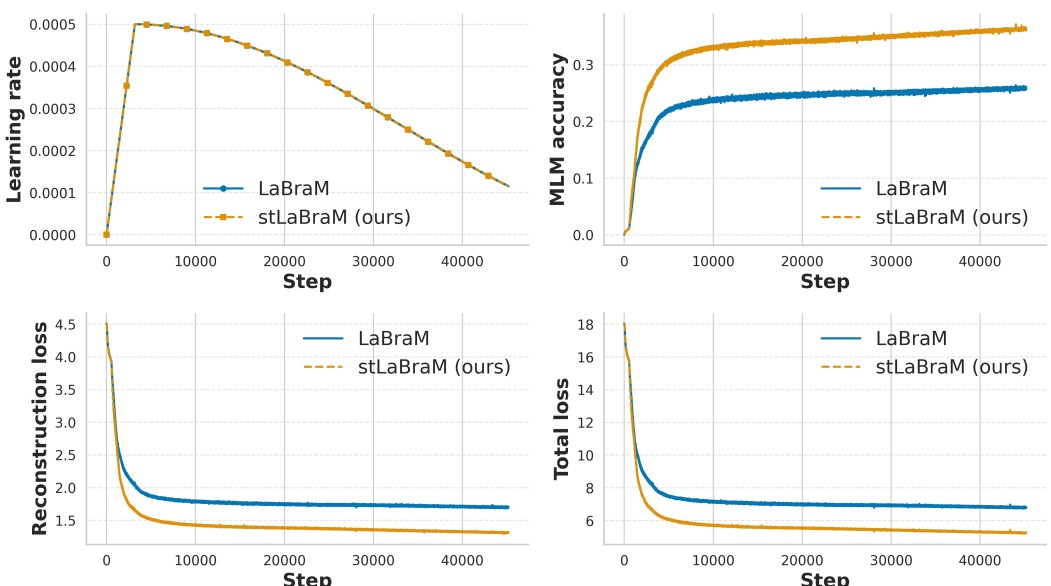

Figure 3: Pre-training results show that stLaBraM achieves lower reconstruction loss and higher MLM accuracy than LaBraM, indicating improved learning dynamics and representation quality from our interpretable spatial-temporal patch embedding.

Table 2: Balanced accuracy for LaBraM and stLaBraM across four benchmark EEG datasets. These benchmarks span a diverse range of real-world EEG scenarios—including clinical (TUAB, ICD-10), event-related abnormality detection (TUEV), and non-clinical cognitive tasks (PhysioNet MI)—thus providing a rigorous evaluation of both in-distribution and challenging OOD settings. stLaBraM consistently outperforms the strong LaBraM baseline across all tasks, clearly demonstrating the broad effectiveness and versatility of our spatial-temporal architecture.

| Dataset | Number of classes | LaBraM (Balanced Acc.) | stLaBraM (ours) (Balanced Acc.) |
|---|---|---|---|
| TUAB (Obeid & Picone, 2016) | 2 | $0.797 \pm 0.004$ | $\mathbf{0.810 \pm 0.004}$ |
| ICD-10 (Appendix A) | 3 | $0.503 \pm 0.008$ | $\mathbf{0.514 \pm 0.008}$ |
| PhysioNet MI (Goldberger et al., 2000) | 4 | $0.471 \pm 0.018$ | $\mathbf{0.634 \pm 0.019}$ |
| TUEV (Obeid & Picone, 2016) | 6 | $0.581 \pm 0.004$ | $\mathbf{0.602 \pm 0.003}$ |

### 5.3 GENERALIZATION, INTERPRETABILITY, AND CLINICAL RELEVANCE

stLaBraM generalized robustly to event and cognitive datasets, outperforming prior foundation models across all domains tested. For Fig. 4, all interpretable block filters are shown—no cherry-picking—and represent aggregate spatial and spectral disparities between classes, consistently aligning with known EEG biomarkers: abnormal runs exhibit high delta, low alpha/beta, matching literature (Buzsáki, 2006). Our method enables direct pattern visualization, supporting rigorous source localization and medical auditability.

Theoretically, the block's effect is grounded in spatial-temporal filter design: under plausible SNR and data/task conditions, spatial-temporal tuning converges to meaningful neural or artifact sources, as analytically motivated and empirically validated here and in prior work (Haufe et al., 2014; Petrosyan et al., 2021). Unlike perturbation-based 'explanations,' our method's patterns are suitable for subsequent inverse modeling.

Figure 4 displays the learned representations on a pair of normal and abnormal EEGs. We show, for each model branch, the difference in spatial patterns $\mathbf{g}_i$ (4a) and the corresponding pair of the power spectral density profiles (4b) in the normal and abnormal conditions. These visualizations high-

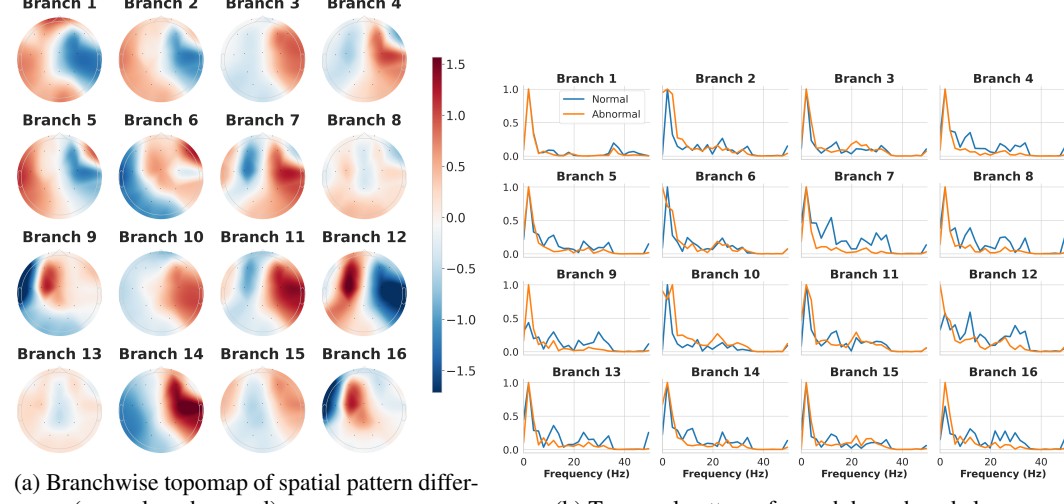

(a) Branchwise topomap of spatial pattern differences (normal − abnormal).

(b) Temporal patterns for each branch and class.

Figure 4: Learned model representations for normal and abnormal EEGs using stLaBraM. (a) Differences in branchwise spatial patterns reveal how the model isolates key discriminative source locations for abnormality; (b) Temporal patterns show class-specific spectral profiles, such as the reduction of alpha/beta activity and increase of delta power in the abnormal condition. These visualizations link model features to neurophysiological signatures.

light features the model finds discriminative for abnormality. Several spatial topography patterns (branches 9, 12, 14, and 16) illustrate model's focusing on the compact frontal neuronal sources. Their corresponding PSDs illustrate that in the abnormal condition the activity in the basic high frequency ranges (alpha: 8-12 Hz, beta: 15-25 Hz) is suppressed while for all mentioned branches lower frequency components (delta: 0.5-3 Hz) exhibit dominating power in the abnormal condition.

These findings by our architecture are in line with the EEG slowing phenomenon that hallmarks various pathological brain conditions, including encephalopathies, traumatic brain injury, and neurodegenerative diseases such as Alzheimer's. It is characterized by increased power in the delta band and decreased power in higher-frequency alpha and beta bands, typically most prominent over frontal and central regions Niedermeyer & da Silva (2005); John (2021); Pucci et al. (1998). This spectral shift reflects widespread cortical dysfunction or disconnection and has been quantitatively confirmed in both clinical and research settings Tolonen et al. (2018); Babiloni et al. (2004). Therefore, after screening the EEG data from the TUAB corpus and training on the downstream classification task, stLaBraM discovered the EEG slowing phenomenon and reached conclusions consistent with the domain-specific literature.

Figure 5 shows topographic clusters collected from the TUAB dataset. We can observe that several clusters (1, 7, 10, 15, 16) are most likely reflecting eye movements activity, while clusters 4, 5, 8, 12, and 14 correspond to fronto-central brain sources. Clusters 5 and 12 of the opposite polarity also exhibit occipital alpha rhythm component known to be one of the dominant features in the EEG. Noteworthy are the clusters 9 and 11 that likely correspond to the cardiac components.

## 6 CONCLUSION

We addressed key limitations in the interpretability and generalization of EEG foundation models by introducing explicit, factorized spatial and temporal filter banks into the patch embedding module of LaBraM. Our approach was evaluated on a comprehensive array of benchmarks—including clinical diagnosis (TUAB, ICD-10), event-related detection (TUEV), and non-clinical cognitive tasks (PhysioNet MI)—covering both in-distribution and out-of-distribution settings. Across all tasks, st-LaBraM delivered consistent accuracy gains, such as a $1.6\%$ relative improvement on TUAB and a $34.6\%$ relative gain on PhysioNet MI, with further substantial error reduction observed in model-

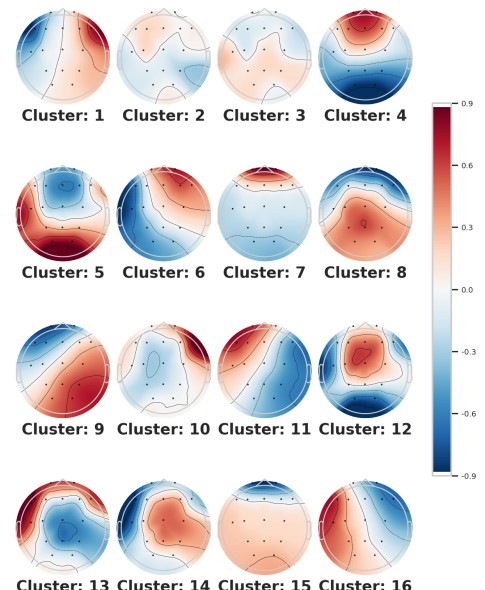

Figure 5: Topography clusters from the TUH dataset, identified via stLaBraM's interpretable spatial filters. Our approach separates neurophysiological patterns (frontal/central/occipital rhythms) from common artifacts (eye movements, cardiac activity), enabling transparent inspection and domain validation of learned EEG representations.

agnostic extensions like stCBraMod (e.g., $43.1\%$ decrease in MSE loss). These advances come with negligible computational overhead and without adding model complexity.

Beyond performance metrics, the spatial-temporal filters learned by our models are directly inspectable and reliably map to neurophysiologically meaningful patterns, such as the characteristic EEG slowing in abnormal conditions. This facilitates neuroscientific insight, supports clinical auditability, and provides a practical alternative to black-box or perturbation-based explainability techniques. Complementary ablation studies confirm that both spatial and temporal components are essential for these gains, and robustness analyses demonstrate that improved representations persist across varied electrode configurations.

Together, these results highlight the critical value of embedding neurophysiological priors in model design—enabling both improved interpretability and robust generalization for clinically relevant EEG applications. Our framework is theoretically grounded, computationally efficient, and compatible with standard clinical EEG protocols, paving the way for transparent, trustworthy deep learning in neuroscience and medicine.

Future work will extend this approach to multi-modal or cross-subject EEG, integrate additional neuroscientific priors, systematically assess interpretability with clinical experts, and explore its synergy with well-established supervised baselines. We hope this work brings the community closer to transparent, robust, and impactful foundation models for neurophysiological and clinical data.

## ETHICS STATEMENT

All EEG data used in this study were collected with informed consent and oversight from Institutional Review Boards (IRBs) or equivalent ethics committees, as applicable. All datasets were fully de-identified prior to use in accordance with relevant privacy and data-protection regulations. The clinically coded dataset (e.g., ICD-10 coded annotations) is not publicly released to protect participant confidentiality; qualified researchers may request access after the review period, subject to appropriate data use agreements and any required IRB approvals. Data processing followed protocols designed to minimize any risk of re-identification or misuse of sensitive information. We acknowledge potential biases of the dataset (e.g., demographic and clinical imbalances) in the Ap-

pendix A. No personally identifiable information was accessed or shared, and all analyses were conducted solely for scientific research. We adhere to the ICLR Code of Ethics, including avoiding discrimination, stigmatization, or harm; potential applications should not be used for clinical decision-making without appropriate validation and oversight.

## REPRODUCIBILITY STATEMENT

We describe in detail model architecture, datasets, and preprocessing as well as provide mathematical formulas for interpretable layers in the main text and provide full training details and hyperparameters in the Appendix B, including hardware specifications, to facilitate independent replication.

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

## A    ICD-10 Dataset Collection and Description

The ICD-10 EEG dataset used in this study was collected from multiple clinical partner sites specifically for the purpose of validating foundation EEG models in out-of-distribution (OOD) scenarios. Detailed characteristics are as follows:

- **Subjects:** About 7,000 unique patients (age range: 4–92, around 60% male), presenting for routine or diagnostic EEG studies.
- **Diagnostics and labeling:** Each subject assigned a primary diagnosis by a licensed neurologist according to ICD-10 codes. For our benchmark, we focus on three groups:
    - F00–F09 (organic mental disorders such as dementia and delirium), 3210 samples;
    - F20–F29 (schizophrenia and related disorders), 2346 samples;
    - F70–F79 (intellectual disabilities), 1219 samples.
- **EEG Acquisition:** All recordings were performed with approval of institutional review boards/ethics committees at each participating center, using standard clinical EEG hardware and protocols. Sampling rates ranged from approximately 200 to 500 Hz, with 16 to 19 channel montages.
- **Preprocessing:** Preprocessing followed the procedures in Section 4 and included channel mapping to the standardized 16-channel montage, artifact rejection, and segmentation.

- **Data Independence:** No subject or recording in our ICD-10 dataset overlaps with the TUH/TUAB corpus or any other public EEG benchmark.

- **Ethics and Consent:** All patients participating in the study provided written informed consent in accordance with institutional guidelines and regulations, and data collection protocols were reviewed and approved by the relevant institutional review boards (IRBs) or ethics committees at each participating clinic.

- **Availability:** The dataset is not publicly downloadable due to clinical privacy restrictions, but may be available to qualified researchers on reasonable request and with appropriate IRB approval.

## B  PRE-TRAINING AND FINE-TUNING PROTOCOLS

Table 3: LaBraM and stLaBraM training hyperparameters for pre-training and fine-tuning.

| Hyperparameter | Pre-training | Fine-tuning |
|---|---|---|
| Batch size | 256 | 64 |
| Peak learning rate | $5 \times 10^{-4}$ | $5 \times 10^{-4}$ |
| Minimal learning rate | $1 \times 10^{-5}$ | $1 \times 10^{-5}$ |
| Learning rate scheduler | Cosine | Cosine |
| Optimizer | AdamW | AdamW |
| Adam $\beta_1, \beta_2$ | (0.9, 0.98) | (0.9, 0.999) |
| $\epsilon$ | $1 \times 10^{-8}$ | $1 \times 10^{-8}$ |
| Weight decay | 0.05 | 0.05 |
| Total epochs | 70 | 60 |
| Warmup epochs | 5 | 5 |
| Gradient clipping | 3.0 | 3.0 |
| Layer scale init | 0.1 | – |
| Hardware (GPUs, memory) | $2 \times$ NVIDIA A100 80GB | $1 \times$ NVIDIA A100 80GB |

Table 4: CBraMod and stCBraMod training hyperparameters for pre-training and fine-tuning, following the same dataset splits and general procedure as for LaBraM.

| Hyperparameter | Pre-training | Fine-tuning |
|---|---|---|
| Batch size | 256 | 64 |
| Peak learning rate | $5 \times 10^{-4}$ | $1 \times 10^{-4}$ |
| Minimal learning rate | $1 \times 10^{-4}$ | $1 \times 10^{-6}$ |
| Learning rate scheduler | Cosine | Cosine |
| Optimizer | AdamW | AdamW |
| Adam $\beta_1, \beta_2$ | (0.9, 0.999) | (0.9, 0.999) |
| $\epsilon$ | $1 \times 10^{-8}$ | $1 \times 10^{-8}$ |
| Weight decay | 0.05 | 0.05 |
| Total epochs | 40 | 60 |
| Warmup epochs | – | – |
| Gradient clipping | 1.0 | 1.0 |
| Hardware (GPUs, memory) | $2 \times$ NVIDIA A100 80GB | $1 \times$ NVIDIA A100 80GB |

This standardized protocol ensures rigorous, reproducible evaluation and fair comparison across foundation model variants and pretraining strategies.

## C  MODEL-AGNOSTIC EVALUATION: STCBRAMOD RESULTS

To further demonstrate the universality and model-agnostic effectiveness of our interpretable spatial-temporal patch embedding, we integrated it into the CBraMod backbone, yielding **stCBraMod**. Both CBraMod and stCBraMod were pre-trained from scratch on the same TUAB splits under the same masked modeling objective as for LaBraM/stLaBraM (see Section B).

**Pre-training dynamics.** stCBraMod consistently exhibited lower masked modeling reconstruction loss and faster convergence than CBraMod:

- At 10 epochs, MSE decreased from 0.00244 (CBraMod) to 0.00165 (stCBraMod), a 32.4% relative reduction.

- At 40 epochs, MSE decreased from 0.00218 to 0.00124, a 43.1% relative reduction.

These trends mirror those observed for stLaBraM, indicating that the proposed spatial-temporal embedding improves representation learning irrespective of the backbone.

**Downstream TUAB abnormal EEG classification.** After fine-tuning, stCBraMod outperforms CBraMod in balanced accuracy (mean±std over $n=3$ random seeds; robustness assessed with 1,000 bootstrap test samples):

- CBraMod: $0.7967 \pm 0.0030$;

- stCBraMod (ours): $0.8147 \pm 0.0034$.

This corresponds to an absolute gain of +0.0180 balanced accuracy and a relative improvement of +2.3% over the baseline, consistent across seeds and robust to resampling.

In summary, the spatial-temporal interpretable patch embedding yields clear benefits in both pre-training loss and downstream performance, supporting its *model-agnostic* utility beyond a single backbone.

## D  ABLATION STUDIES

We ablate the spatial and temporal filtering modules and assess robustness to sensor corruption. We report balanced accuracy (mean ± std) on the TUAB Abnormal EEG Corpus (Obeid & Picone, 2016).

**Spatial–temporal module contribution.** The spatial-only variant provides negligible improvement over the baseline, whereas the full spatial–temporal block delivers a clear, statistically robust gain (Table 5). Relative to the baseline, the full block improves balanced accuracy by +0.013 absolute (from $0.797$ to $0.810$), corresponding to a +1.63% relative increase and an effect size exceeding $3\times$ the reported standard deviation. In contrast, removing temporal filtering yields only +0.001 absolute (+0.13% relative), indicating that joint spatial–temporal modeling is essential.

Table 5: Ablation of spatial/temporal modules on TUAB abnormal EEG classification. $\Delta$ columns are absolute and relative change vs. the baseline.

| Variant | Balanced Acc. | Abs. $\Delta$ vs. base | Rel. $\Delta$ vs. base |
|---|---|---|---|
| No interpretable block (baseline) | $0.797 \pm 0.004$ | – | – |
| Spatial-only (no temporal filtering) | $0.798 \pm 0.004$ | $+0.001$ | $+0.13\%$ |
| **Full spatial–temporal block (ours)** | **$0.810 \pm 0.004$** | **$+0.013$** | **$+1.63\%$** |

**Robustness to sensor corruption (channel masking).** We quantify sensitivity to physiologically meaningful sensor locations by masking channels (Table 6). Masking frontal electrodes (FP1, FP2, F7, F8, FZ) causes a drop of $-0.068$ absolute (from $0.810$ to $0.742$), i.e., $-8.40\%$ relative, suggesting the model leverages frontal activity relevant to the task. Randomly masking half the channels results in a $-0.045$ absolute ($-5.56\%$ relative) decrease, indicating reasonable robustness yet clear dependence on sensor availability.

Table 6: Robustness to channel masking on TUAB. $\Delta$ columns report change relative to the full 16-channel model.

| Masking scenario | Balanced Acc. | Abs. $\Delta$ vs. full | Rel. $\Delta$ vs. full |
|---|---|---|---|
| All 16 channels (full model) | $0.810 \pm 0.003$ | – | – |
| Mask frontal (FP1, FP2, F7, F8, FZ) | $0.742 \pm 0.003$ | $-0.068$ | $-8.40\%$ |
| Mask 8 random channels | $0.765 \pm 0.006$ | $-0.045$ | $-5.56\%$ |

Taken together, these results indicate that the interpretable front end relies on physiologically grounded features—particularly frontal activity relevant to abnormality—while degrading gracefully under partial channel availability. This aligns with clinical knowledge and supports the validity of the learned spatial filters. Practically, it suggests stLaBraM can operate across heterogeneous or partially missing montages with modest performance loss, and motivates future work on channel-agnostic training (e.g., sensor dropout, montage harmonization) and source-space alignment to further improve robustness.

