# OpenReview forum: "stLaBraM: Interpretable EEG Foundation Models with Factorized Spatial-Temporal Patch Embedding for Self-Supervised Learning"
_ICLR.cc/2026/Conference — Submitted to ICLR 2026_

### Official Review · Reviewer_Mh4k · 2025-10-20

**Soundness:** 3
**Presentation:** 2
**Contribution:** 2
**Rating:** 4
**Confidence:** 4

**Summary:**

This paper proposes stLaBraM, an interpretable EEG foundation model that introduces a factorized spatial–temporal patch embedding to improve both interpretability and performance in self-supervised pretraining. The proposed module replaces the standard patch embedder of LaBraM with learnable spatial filters (for channel unmixing) and temporal filters (initialized on canonical EEG frequency bands), allowing each latent branch to correspond to meaningful neurophysiological sources and oscillatory components.

**Strengths:**

- Interpretability of EEG foundation models is indeed a critical and underexplored topic. The paper's motivatio, combining high performance with neuroscientific interpretabilit, is strong and well-justified.
- The proposed interpretable patch embedding design is elegant and principled. It leverages classical signal processing theory (factorized spatial–temporal filtering) to produce interpretable filters that can be mapped to cortical sources and spectral profiles.
- Visualization of spatial topographies and frequency responses aligns well with established EEG phenomena such as alpha suppression and delta enhancement in abnormal conditions. This significantly strengthens the claims of neurophysiological interpretability.
- The authors systematically ablate spatial vs. temporal components and test robustness under channel masking, showing that the method degrades gracefully and relies on physiologically meaningful sensors (e.g., frontal electrodes).

**Weaknesses:**

- Figure 2 (architecture diagram) is low-resolution and somewhat cluttered. A vector graphic version with clearer labeling of the spatial–temporal filtering stages would improve readability.
- The model uses a fixed 16-channel montage during both pretraining and finetuning. It is unclear whether the proposed patch embedding can handle arbitrary or partially missing electrode configurations (e.g., via sensor dropout or interpolation). Clarifying this limitation or extending to montage-agnostic training would enhance generalizability.
- While the four benchmarks cover several clinical and cognitive domains, additional representative EEG tasks such as emotion recognition or sleep stage classification could further validate the model's universality.
- The interpretability claim is primarily qualitative. Quantitative validation (e.g., correlation with known EEG frequency bands or expert ratings of spatial maps) would make the claim more rigorous.

**Questions:**

- The proposed spatial–temporal filters may introduce data leakage between patches by allowing information exchange across channels and time windows. Could this make the masked modeling task artificially easier and thereby explain part of the improved reconstruction loss? How did you ensure a fair comparison with the baseline LaBraM?
- How well does the method generalize to EEG data with different electrode configurations or missing sensors? Can the interpretable embedding be adapted to variable montages without retraining from scratch?

---

### Official Review · Reviewer_Mib4 · 2025-10-26

**Soundness:** 2
**Presentation:** 2
**Contribution:** 1
**Rating:** 2
**Confidence:** 5

**Summary:**

The paper proposes stLaBraM, whose core contribution is a novel, neuroscience-prior-based factorized spatial-temporal patch embedding module. This module decouples spatial filtering and temporal filtering, making the model interpretable by design. Experiments demonstrate that this interpretable design not only avoids a performance trade-off but actually improves performance over the LaBraM baseline. The model's interpretability is validated by its ability to automatically discover clinically meaningful biomarkers, such as the EEG slowing phenomenon, from data.

**Strengths:**

1. The target issues of the paper are meaningful and worth exploring and the motivation is clear.
2. The paper is well written and easy to follow.
3. The concept of decomposing the embedding layer into spatial and temporal filters is based on the physical model of EEG signal generation, providing a neuroscientific rationale.

**Weaknesses:**

1. The most severe issue lies in the paper's claim of being a “foundational model” and “SOTA”, while completely avoiding comparison with current SOTA models in its experiments. Despite mentioning models like BrainWave in the introduction, Table 2 completely omits these models, let alone numerous others such as LaBraM++, CBraMod, NeuroLM, EEGFormer, EEGMamba, Neuro-GPT, and SPICED. The paper's comparative experiments are severely flawed.
2. The downstream tasks are insufficient. The authors only conduct experiments on 4 downstream tasks, can authors conduct more?
3. The contribution of this work is incremental. Even though in the Appendix the authors conduct experiments on other CBraMod, the novelty is still limited and the related analysis and experiments are insufficient. The authors claim their module is “model-agnostic”, yet the evidence consists solely of its application to a single model, CBraMod. This is wholly inadequate to demonstrate its universality.
4. The paper's performance improvements are highly questionable. It achieves a mere 1.6% improvement on the core clinical task TUAB, yet a staggering 34.6% gain on PhysioNet MI. This strongly suggests the module's “improvements” are not generalizable but likely overfit to specific task characteristics, contradicting its “base model” generalization claims.

**Questions:**

1. It seems that the authors do not describe the usage of LLMs in the paper. This constitutes a violation of this year's ICLR regulations.
2. See weakness

---

### Official Review · Reviewer_R1nN · 2025-11-01

**Soundness:** 3
**Presentation:** 3
**Contribution:** 3
**Rating:** 6
**Confidence:** 4

**Summary:**

This work proposes stLaBraM, which is an EEG brain foundation model that makes LaBraM's front end explicitly interpretable by factorizing patch embedding into learnable spatial virtual channels and band-specific temporal filters. The design ties tokens to real neurophysiology, and a Haufe-style pipeline maps learned weights to scalp topographies and power spectra for source-level insight. The result is a compute-neutral module that improves masked-modelling and transfers better across clinical and cognitive tasks. The ablations of the work show both spatial and temporal filters both necessary, and the robustness tests show noticeable degradation under channel masking, and the learned patterns recover known EEG slowing in abnormal cases, supporting clinical validity. Overall, from my understanding, this foundation model advances the EEG foundation model by pairing stronger performance with transparent and source-localizable features that clinicians can inspect.

**Strengths:**

I found the originality, clarity, quality, and significance of the work very well. The work's core idea is original, as it makes the patch embedding of an EEG foundation model explicitly interpretable by factorizing it into learnable spatial virtual channels and band-targeted temporal FIR filters, then mapping the learned weights to scalp topographies and power spectra with a Haufe-style pipeline. This is a creative, low-cost way to ground tokens in real neurophysiology and remove a key limitation of prior FMs. I also found the quality of the empirical part very well and easy-to-follow, as the pretraining shows clear gains in MLM accuracy and reconstruction loss with almost no extra compute, and the same idea helps another backbone, which indicates model-agnostic value, and I think it'd be good to know that. The significance is also compelling, as the model pairs accuracy gains with source-level insight, recovers known clinical patterns like EEG slowing, and is compatible with standard EEG workflows, which meaningfully advances trustworthy EEG FMs for clinical and scientific use.

**Weaknesses:**

While I found the method interesting, I have a couple of concerns that I'd like to raise. The method and claim of interpretability come from your factorized spatial-temporal filters and the Haufe-style mapping. I was wondering if it would be possible for the authors to add direct head-to-head baselines that use classic, simple EEG front-ends (e.g., spatial mixing + band filters) under the same pretrain/fine-tune setup? In this case, we'd be able to see the gain beyond a strong, simple baseline, and it could be useful to see it.

Also, the pretraining uses a masked position objective, which is ok, but I think it leaves open whether your front end helps under other Self-Supervised Learning (SSL) losses, so I'm curious to see the repeat key ablation with a masked regression or contrastive loss to show the benefit is not tied to this single objective. If possible, I'd encourage the authors to do this ablation and share the results here or at least share their thoughts on it, and please note that this is not a strong weakness, so there won't be any score deduction because of this point.

Another point that I would like to make is that most downstream results keep the patch embedder frozen. That choice protects interpretability, but it may limit accuracy or bias the comparison. I was wondering if the authors have any thoughts on it. Maybe a small 2*2 study (frozen vs end-to-end) would clarify the trade-off.

Finally, the pipeline filters to recordings with all 16 channels, and then applies band-pass/notch filters, and uses fixed 16-second windows. I was wondering if the authors have already tested across montages (e.g., train on 16-ch, and the test on 19-ch mapping), and varied window length? Maybe sharing a leave-center-out result to show robustness beyond these choices could make it clearer.

**Questions:**

I'd encourage the authors to check the weaknesses part first. For the questions, I have two more questions from the authors and would appreciate their thoughts/comments.

(1) I was wondering if the authors could show/share that the filters are the cause of the gains by swapping in random spatial or random temporal filters while keeping everything else fixed, and measuring the drop for each?

(2) I'm curious to know what the out-of-domain generalization is if you train on TUAB only and test on TUEV and PhysioNet MI with no finetuning, and if you train on one center and test on a held-out center or on a 19-channel montage? Asking this because clinical value depends on robustness, and this kind of experiment checks whether your interpretable front end truly improves out-of-domain generalization across datasets, centers, and montages rather than overfitting one setup. If the authors could provide their thoughts on it only, that'd make sense as well.

---

### Official Review · Reviewer_qgMe · 2025-11-01

**Soundness:** 3
**Presentation:** 3
**Contribution:** 2
**Rating:** 2
**Confidence:** 3

**Summary:**

The paper proposes stLaBraM, a modification of LaBraM in which the patch embedder is factorized into a learnable spatial unmixing that produces virtual channels and a bank of temporal FIR filters intended to align with canonical EEG bands. The authors derive the Haufe-style pattern estimates for spatial maps and frequency profiles and use these for visual interpretability as well. The authors report higher MLM accuracy and lower reconstruction loss than LaBraM. On downstream tasks, stLaBraM yields small gains over their LaBraM implementation and similar model agnostic gains are shown for CBraMod.

**Strengths:**

* Clear motivation and well-posed problem: Interpretable components to EEG foundation models is important and clinically relevant. the paper explains why factorized spatial temporal filters can be mapped to physiologically meaningful sources.
* Architectural Scalability: The components added have 0 parameters and negligible training time relative to LaBraM making the additions simple and scalable.
* Derivation: The Haufe-style formulation is impressive and the visualization pipeline is sound.
* Empirical Improvments: Gains in pretraining and heldout performance are seen although not consistently.

**Weaknesses:**

I think some claims are are little far reaching and have some concerns about test-overlap. Please see below.

* SOTA Claims and Baselines: The SOTA claims, to me, aren't entirely backed up. The quantitative comparisons are only with LaBraM and CBraMod and that too, these are only author implementations. There are no comparisons to strong and widely used EEG Baselines like EEGNet, BrainWave, BRANT, etc on the same preprocessing/splits. I would not say that the SOTA claim is fully substantiated and I would like to see some of these baselines added.
* Evaluation: One concern I have is whether there is any overlap between the pretraining corpus and test corpus or whether improvements could be explained by such overlap. The paper's pretraining corpus is the TUH EEG Corpus while downstream includes TUAB and TUEV. The paper does not state that the TUAB/TUEV subjects were removed from the TUH pretraining pool, which LaBraM does upon further reading. Can the authors please confirm this? If this is not removed, I would not call the evaluation on TUAB and TUEV OOD evaluations.
* Interpretability: The interpretability evaluations mostly use visual inspection and some statements seem like value judgements to me like "most likely reflecting eye movements". This is done without external validation or analyses across seeds/sessions/sites. The paper does not actually perform source localization.
* Ablations: The spatial-only ablation shows no improvement while the full block shows some improvement. There is no temporal only ablation nor sensitivity to K or filter length. These are important for interpretability. I think this makes it more challenging to understand whether the spatial aspect of the method is clearly necessary for imporvements.

**Questions:**

* Patch-Embeds: I was a bit confused about section 3. It states that the paper "applies a bank of K 1D temporal filters to each virtual channel" but the formula doesn't state this. It seems to use a one-to-one pairing in $r$ if I am understanding correctly. The text later claims that stacking K yields band-filtered time series. It is unclear whether temporal filters are shared across channels, paired one-to-one, ior applied as a full bank per channel. Could the authors clarify?
* Reconstruction loss:Why is the reconstruction loss written as a cross-entropy over patch-positions rather than content reconstruction? Why would you predict indices instead of some MAE-style objective?

---

### Meta-Review · Area_Chair_FiTY · 2026-01-08

**Summary:**

This paper proposes an EEG foundation model.

The reviewers agreed that the paper is on a relevant and interesting topic, and that the paper is clearly written with respect to the work performed.

All four reviewers seem to be displeased by the qualitative nature of the interpretable features. To the authors' credit, it is not clear what other form interpretation can come in, but I generally agree that this claim is weak, and that visual inspection of signals and weights otherwise described as complex is somewhat suspect.

Two reviewers dispute SOTA claims (`qgMe` and `Mib4`). Another reviewer finds the opposite, that the empirical results section is "very well" (`R1nN`). I tend to agree with `qgMe`, who lists specific baselines with open implementations.

As a technicality, please be aware of `Mib4`'s comment that this year's ICLR (and likely future ICLRs) will require LLM usage descriptions.

**Reviewer Concerns:**

No rebuttal addressing concerns.

**Reviewer Scores:**

No change.

---

### Decision · Program_Chairs · 2026-01-26

Reject